# Liquid Biopsies, Novel Approaches and Future Directions

**DOI:** 10.3390/cancers15051579

**Published:** 2023-03-03

**Authors:** Athanasios Armakolas, Maria Kotsari, John Koskinas

**Affiliations:** 1Physiology Laboratory, Medical School, National and Kapodistrian University of Athens, 115 27 Athens, Greece; 2B’ Department of Medicine, Hippokration Hospital, National and Kapodistrian University of Athens, 115 27 Athens, Greece

**Keywords:** cancer, diagnosis, prognosis, CTCs, ctDNA, miRNA, proteome, exosomes, clinical applications

## Abstract

**Simple Summary:**

The gold standard for detecting cancer and profiling tumors is tissue biopsies. Despite this, tissue biopsies have been associated with many limitations leading to the desire for less invasive and more accurate solutions. One very attractive candidate for the diagnosis and prognosis of cancer in patients is provided by liquid biopsies. The number of analytes circulating in the blood that may be used for liquid biopsy testing is enormous making it a promising technique for the clinical management of oncological patients. The goal of this study is to discuss in detail the clinical relevance of liquid biopsies, as well as the opportunities they might offer for cancer prognosis, diagnosis and monitoring. The isolation process and clinical use of the biological components of the liquid biopsy will also be explained, with specific focus placed on novel procedures that can be developed as well as the approach’s future possibilities.

**Abstract:**

Cancer is among the leading causes of death worldwide. Early diagnosis and prognosis are vital to improve patients’ outcomes. The gold standard of tumor characterization leading to tumor diagnosis and prognosis is tissue biopsy. Amongst the constraints of tissue biopsy collection is the sampling frequency and the incomplete representation of the entire tumor bulk. Liquid biopsy approaches, including the analysis of circulating tumor cells (CTCs), circulating tumor DNA (ctDNA), circulating miRNAs, and tumor-derived extracellular vesicles (EVs), as well as certain protein signatures that are released in the circulation from primary tumors and their metastatic sites, present a promising and more potent candidate for patient diagnosis and follow up monitoring. The minimally invasive nature of liquid biopsies, allowing frequent collection, can be used in the monitoring of therapy response in real time, allowing the development of novel approaches in the therapeutic management of cancer patients. In this review we will describe recent advances in the field of liquid biopsy markers focusing on their advantages and disadvantages.

## 1. Introduction

As cancer continues to be one of the leading causes of death worldwide, continuous efforts are being made to diagnose and manage this disease. Although, tissue biopsies have been the most common methods for diagnosing cancer and profiling the tumor, they are associated with many limitations [1].

Typically, tissue biopsies are an invasive method and for some anatomical sites it is not easy to collect them. They also provide a limited picture for intratumoral and intermetastatic genetic heterogeneity, as tumors are heterogeneous entities containing various subpopulations of cells that feature different lesions [1,2]. Furthermore, cancer cells over time undergo genetic and epigenetic changes and can evolve dynamically, guided by microenvironmental stimuli and clonal selection due to therapy pressure. This results in further tumoral heterogeneity [1], thus affecting the accuracy of the examination and the therapeutic decisions made based on it. In addition, surgical biopsies have limitations in terms of time, repeatability, age of the patient, cost and sometimes can even cause harmful clinical complications [3]. Therefore, they are not suitable to highlight the overall tumor profile, to identify any lesions in different locations nor to be used for the longitudinal monitoring of the disease [4].

The solution to the above issues comes from liquid biopsies, which constantly gain ground in terms of prognosis, diagnosis and monitoring of the progression of the disease. This method offers the advantage of a less invasive nature, lower cost, real-time information on the state of the tumor and, in some cases, the ability to overcome the issue of tumor heterogeneity (or multiple metastatic alterations) [3]. Such biopsies include sampling and analysis of body fluids, usually blood, although other sources such as urine, saliva, cerebrospinal fluid (CSF) and bone marrow can be used [5].

Biologically, the targets for liquid biopsy can be divided into two categories. One category refers to large or small molecules without cells or without a subcellular structure in the body fluid; these include proteins, nucleic acids, lipids, carbohydrates and other small metabolites and metal ions. The second category includes targets with cellular or subcellular structures, including single or clustered circulating tumor cells (CTCs), circulating cancer-related fibroblasts (CAF), immune cells, tumor-educated platelets (TEP) [6], extracellular vesicles (EVs) and circulating mitochondria [7,8]. Recent evidence suggests that exosomes operate on numerous receptor cells via a range of bioactive chemicals in vesicles and play a significant role in immune surveillance, angiogenesis, tumor formation, metabolism and inflammatory responses [9]. However to date, only circulating tumor DNA (ctDNA) and circulating tumor cells (CTCs) are the components whose clinical application has been approved by the US Food and Drug Administration (FDA) [2].

The purpose of this review is to describe in detail the clinical significance of liquid biopsies as well as the possibilities it can offer to the prognosis, diagnosis and monitoring of cancer progression. The isolation method and the usefulness in clinical practice of the biological components of a liquid biopsy will be described as well, and special emphasis will be given to the new techniques that can be developed as well as to the future prospects of the method.

## 2. CTCs

Metastasis is a multi-step process that depends on the presence of CTCs in the blood stream and/or disseminated tumor cells (DTCs) that are found in the bone marrow [7,8]. In order for CTCs to be able to disseminate from primary tumors they must undergo phenotypic changes that will allow the cells to penetrate blood vessels [10]. The epithelial-mesenchymal transition (EMT) is a central process in metastasis where the cancer epithelial cells downregulate the expression of their epithelial markers, including the cell membrane proteins that are responsible for cell to cell adhesion, and they express the mesenchymal markers [11,12,13]. Mesenchymal cells do not possess cell adhesion molecules on their surface and, therefore, can easily detach from the main tumor. In addition these cells induce the expression of proteases and integrins that are central molecules in the intravasation and extravasation of mesenchymal cancer cells [14,15]. CTCs are a population of cancer cells that manage to detach from either the primary tumor or metastatic deposits in the periphery of patients, and they seem to have a short half-life of approximately 1h to 2.4 h. The presence of CTCs in the bloodstream consists of a very heterogenous population that varies greatly in number from patient to patient and even within the same patient at different time points.

The presence of CTCs in the circulation is fundamental for the development of metastasis in various types of solid tumors [16,17] (Figure 1). The fact that CTCs are highly heterogeneous and circulate in low numbers renders them a very hard target to detect accurately enough to set the guidelines for patient treatment (Figure 1). Therefore, it can easily be understood that the first step of enrichment is critical for the analysis of the cancer cell load (metastatic or not) in the periphery of the patients. For this reason, a variety of techniques have been developed, based on both the biophysical and biological properties of these cells, in order to differentiate them from their background and enrich them, so that they are compatible with molecular analysis or imaging analysis. Similarly, many technologies have been developed for the capture, isolation and detection of CTCs [18,19,20,21,22].

### 2.1. Enrichment and Isolation of CTCs

Due to the small CTC concentration in the blood, analysis always starts with an enrichment step that aims to increase the concentration of these cells by several logarithmic units thus allowing an easier identification of single tumor cells. CTCs can be enriched by approaches that exploit differences between tumor and normal blood cells, based on biological properties such as the differential expression of protein markers or different physical properties of cells including size, density, deformability or electric charges, and these enrichment principles can be combined to optimize the yield of CTCs [17,24,25].

A variety of devices has been developed to enrich and detect CTCs, with emphasis on devices capable of selecting and detecting CTCs that have undergone EMT [25,26]. The EPCAM-based enrichment for CTC detection has provided a reliable prognostic tool in different carcinomas [27,28]. However other epithelial cell surface antigens including EGFR7 [29] and mucin 1 [30], and tissue specific antigens such as prostate specific membrane antigen (PSMA) [31] for prostate cancer cells and ERBB2 [32] for breast cancer cells have been exploited for this purpose (Table 1). CTCs present a very heterogeneous population of cells. Recent evidence indicates that in many cases CTCs could cease to express the selected marker, leading to markers escaping detection and, thus, to false negative results [33,34]. Consequently, the bias which might be introduced by positive selection can be avoided by negative selection. In this case non-malignant blood cells are depleted from the blood using antibodies that recognize the cell surface antigens expressed on leukocytes, usually CD45 and other cells in the bloodstream, including endothelial stem cells with markers such as CD146 and hematopoietic stem cells with markers such as CD34 [35,36]. Disadvantages of negative selection include the lower purity of isolated CTC populations compared to the techniques of positive selection and the risk of CTCs becoming trapped in a mass of blood cells and, thus, being included in the depleted cell fraction and ignored [37,38,39].

With regard to techniques based on the physical differences of tumor cells and non-malignant blood cells, it is worth noting that these characteristics are highly variable between CTCs and have substantial overlap with those of non-malignant cells; therefore, definitions of CTC size depend on the capture device. Microfiltration technologies have been developed where blood is passed through pores or microfluidic passageways with calibrated size to trap CTCs, resulting in size exclusion and, therefore, retention of large CTCs, albeit with possible loss of small CTCs [40]. Other microfluidic devices that rely on size separation use inertial focusing strategies to separate CTCs from other blood components, while dielectrophoresis (DEP) allows the separation of CTCs based on the different electrical charges of tumor and blood cells. Special microfiltration systems have also been developed to specifically capture CTC clusters based on size exclusion [41]. Most CTCs occur as single cells, but CTC clusters can be detected and their biology is still being investigated [42].

**Table 1 cancers-15-01579-t001:** Clinical applications of CTCs.

Marker	Assay Relevance	Disease	Technique	Advantages	Disadvantages	References
ERBB2	Prognostic/guiding therapy	Breast cancer	CellSearch	First-line ERBB2-targeted therapy for metastatic breast cancer appears to reduce CTC levels more than endocrine or chemotherapeutic therapy; anti-ERBB2 therapy appears to be linked with lower total CTC levels.	Retrospective analysis. The small number of patients with progressive disease highlights the implicit difficulties in analyzing the rate of CTC-positive cases.	[43]
PDL-1	Guiding therapy	Breast cancer	CellSearch	CTC and platelet PD-L1 expression might be used to determine which patients should be treated with immune checkpoint inhibitors and as a pharmacodynamic biomarker during therapy.	The CellSearch platform might potentially disrupt CTC doublets and clusters, underestimating their biological/prognostic effect compared with other assays.	[44]
EpCAM	Prognostic	Metastatic breast cancer	CellSearch	CTC counts as independent prognostic factor. CTC testing is more repeatable than radiology and detects progression earlier.	The findings of the study do not support the use of the test as a screening tool for detecting new primary or metastatic breast cancer.	[45,46,47]
		Nonmetastatic primary breast cancer	Laser scanning cytometry	A rise in CTC counts of more than ten-fold at the conclusion of therapy is significantly predictive of recurrence.	The relationship among CTC number and therapeutic efficacy may change amongst patients.	[48]
			CellSearch	CTCs are considered an independent prognostic factor.	CTC detected only in 24% of patients studied.	[49,50]
		Non-small-cell lung cancer	CellSearch	CTC reduction as an early indicator of therapeutic response.	Biomarkers’ predictive or prognostic value cannot be differentiated.	[51]
		Small-cell lung cancer	CellSearch and ISET	CTC isolation using ISET is dependent on cellular size and independent of any cellular marker.	NA	[52]
		Metastatic colorectal cancer	CellSearch	CTC count as independent prognostic factor.	Overall CTC yield is less than in other epithelial malignancies such as breast cancer.	[53]
		Non metastatic colorectal cancer	CellSearch	CTC count as independent prognostic factor.	It is uncertain if CTCs discovered are precursors of metastatic lesions or if CTCs arise from metastases and are just a marker of overall disease burden.	[54]
		Hepatocellural cancer	CellSearch and EpCAM-based immunoenrichment and FACS	The study demonstrated the feasibility of utilizing CTC-derived DNA for next-generation sequencing.	NA	[55]
		Castration-resistant prostate cancer	CellSearch	At all time points, CTC numbers predicted OS better than PSA decrement methods.	NA	[56,57]
ALK	Predictive	Lung adenocarcinoma	ISET-ICC/FISH	Noninvasive real-time monitoring of targeted treatment is a possibility.	NA	[58]
ASGR1	Prognostic	Hepatocellular carcinoma	Antibody-coated magnetic beads based separation	In terms of specificity and sensitivity, CTC detection exceeds AFP mRNA.	Comparison with other studies is problematic owing to the unique technique utilized.	[59]
PSA, PSMA, PSCA, KRT19	Prognostic	Prostate cancer	RT-PCR	High sensitivity of RT-PCR.	False positive and false negative results are possible.	[60]
Pan-cytokeratin, AR-V7, CD45	Prognostic/predictive	Metastatic castration-resistant prostate carcinoma	AdnaTest	Taxanes as chemotherapeutic drugs of choice for individuals with androgen receptor signaling blocker resistance (based on CTC AR-V7 positivity).	Due to the small sample size, multivariable analysis to connect AR-V7 status with prognosis and define subpopulations was not possible.	[61]
			Epic AR-V7 Test	Stratification for CTH based on taxanes in mCRPC CTC AR-V7+.	NA	[62,63]
Tyrosinase	Prognostic	Malignant melanoma	RT-PCR	CTC count as independent prognostic factor.	NA	[64]
MART1, MAGE-A3, and PAX3	Prognostic	Malignant melanoma	RT-PCR	CTC count as independent prognostic factor.	The presence of CTCs was not related to any of the identified clinical prognostic indicators.	[65]

### 2.2. Detection of CTCs

After enrichment, an identification step is required to detect CTCs surrounded by residual leukocytes at the single cell level by immunological, molecular or functional methods [66]. The dominant methods use antibodies against membrane and cytoplasmic antigens, including epithelial, mesenchymal, histospecific and tumor-related markers, with the aim of direct immunological detection [26]. Until now, the only clinical application of CTCs approved by the FDA is the CellSearch platform [28] and the most current CTC assays use the same identification step as this one (Table 1). Cells stained with fluorescently labeled antibodies to epithelial cytokeratin (CK) are visualized through fluorescence microscopy and used as a marker of CTCs, while staining of CD45 is used to exclude leukocytes [67]. Some of the markers used vary in different types of cancer, for example cytokeratins apply to breast, colon and prostate cancers, and other epithelial tumors, although specific tissue antigens can also be used, such as prostate specific antigen (PSA) or breast specific mammaglobin [17]. However, this technology has some limitations. First of all, it is mainly based on the expression of EpCAM which has been associated with localized cancer, but during metastasis its expression, along with that of CK, decreases amid the appearance of mesenchymal markers [68]. Second, cell isolation through the CellSearch system is followed by cell fixation for stabilization, which prevents further characterization of viable cells such as CTC cultures, while having a low sensitivity for CTC detection (one cell per 1 mL of blood sample). Finally, the CellSearch system offers low purity of captured cells, in the range of 60–70%, resulting in captured CTCs that are usually contaminated by blood cells or normally circulating epithelial cells (CECs) [42,65,69].

It is possible to manually isolate the identified CTCs by micromanipulation; however, it is laborious and time-consuming. An alternative approach for separation of CTCs in order to further genomic, molecular or functional analyses involves automated selection of single cells using a DEPArray, a device that allows trapping single CTCs in DEPcages [70]. Dielectrophoresis (DEP) is a liquid biopsy separation assay that is based on particles with different polarizations and that move differently under a non-uniform electric field [71]. Microchips that use the DEP method have multiple integrated electrodes which generate the non-uniform electric field in order to isolate and capture single CTCs. However, the low sample volumes and the varying dielectric features of cells due to ion leakage could limit the isolation time [72].

A new fluorescence-activated cell sorting (FACS) approach has also been used for CTC detection and phenotypic analysis, but this technology typically requires a pre-enrichment step to achieve sufficiently high initial CTC concentrations [73]. FACS is a cell-based analytic method where an immunomagnetically enriched blood sample is injected into a fluid stream, and single cells in the stream are interrogated by lasers as they flow into a capillary tube. The cells are then sorted based on light scattering and fluorescence patterns by comparison with negative (healthy blood cells) and positive control (EpCAM-expressing cancer cell lines) [74]. On the other hand, there are some restrictions to this method. Τhe use of expensive antibodies leads to high detection costs, whereas, in many cases CTCs cannot be further analyzed in real-time conditions since the cells are fixed or lysed during the assay process [75,76].

CTCs can also be detected by techniques that target the mRNA or DNA level. These techniques require the design of PCR tests with specific primers for tissue, organ, tumor-specific transcriptions, or for tumor-specific mutations, translocations or methylation patterns unique to the tumor [77]. Furthermore, these technologies also allow the quantification of CTC numbers. Reverse transcription PCR (RT-PCR) assays are the most user-friendly method for detecting low-abundance mRNA transcriptions. A limitation of this approach is that the CTC number can only be estimated due to the fact that gene expression levels vary between CTCs [78]. Currently, digital droplet PCR (ddPCR) allows detection and absolute quantification of low-abundance targets in shorter times without requiring a large number of replications [79,80]. ddPCR relies on water-oil emulsion droplet technology. In comparison with other digital PCR assays, this method has a lower sample requirement, thereby reducing costs and preserving valuable samples [81].

Functional assays like the epithelial ImmunoSPOT (EPISPOT) assay have been used for CTC in vitro detection in blood and bone marrow samples for more than two decades and have been validated at the clinical level for several different cancers [82]. This assay provides quantitative information about the number of viable CTCs present in the sample based on the fluorescence detection of specific epithelial proteins secreted by these cells, as well as qualitative information about which of these proteins are shed during cell culture. Currently, this technique has been further developed allowing for the capture and detection of CTCs at the single cell level. The so-called EPIDROP, as an abbreviation of ELISPOT in a drop, is a more rapid and sensitive form than the previous one [83]. In this assay, CTCs are immunostained prior to individual encapsulation in fluid microdroplets and, consequently, both the total number of CTCs (EPCAM^+^ or EPCAM^−^) and the number of functional CTCs can be imprinted. Indeed, viable CTCs can be distinguished from apoptotic CTCs, and EPCAM^+^ versus EPCAM^−^ CTCs enable the assessment of EMT status. In the future, a subsequent molecular characterization of the captured CTCs will be incorporated into this innovative assay. Despite the fact that EPISPOT is a promising technique, there can be problems when antigen levels are lower or binding efficiency is reduced [84]. Furthermore, processing a single sample in an EPISPOT assay requires three days for analysis [85]. This, in combination with the finding that it may fail to isolate more heterogeneous cells because of its biomarker dependence [86], render EPISPOT unsuitable for clinical use.

### 2.3. Characterization of CTCs and Analytical Technologies

Aiming towards personalized cancer treatment, many innovative technologies have been developed in recent years intended for the characterization of CTCs. CTCs can be analyzed by cytogenetic analyses, such as in situ fluorescence hybridization (FISH), to identify chromosomal rearrangements [87] (Table 1). Multi-omics techniques have entered dynamically in the patient management of cancer. CTC single cell analysis is a novel approach where CTCs are isolated and the entire genome can be amplified in order to make subsequent assessments of duplicate number aberrations and specific mutations using array competitive genome hybridization or next-generation sequencing (NGS) techniques [88]. A physical disadvantage of this method is that the findings cannot be verified because single CTCs are found in limited quantities [89]. Though, the DNA amplification protocol requires careful technical validation to avoid false findings and, thus, ensures a low error rate [62]. In addition, strict bioinformatic approaches are needed to ensure reliable identification of tumor-specific changes in individuals’ CTCs. Therefore, it is a time-consuming technique that also involves high costs [90] (Figure 1). On the other hand, this method offers high sensitivity of CTCs from many tumor types, and the variety of selection markers allows for the possibility of characterizing cells for multiple markers all at the same time [91]. In addition, the use of NGS in CTC analysis offers the possibility of using genomic and transcriptional CTC profiles to improve the understanding of cancer heterogeneity [92].

RT-PCR transcription assays that are not expressed in non-malignant blood cells such as those encoding PSA or epithelial cytokeratins are sensitive enough to allow the detection of single CTCs but can also provide information on their phenotype (Table 1). However, special manipulations are required because low-level external expression of target transcript in infected leukocytes (or other non-malignant cells in the bloodstream) can lead to incorrect attribution of the results (false positives). As an alternative approach, the transcriptomic profile of single CTCs isolated by micromanipulation can be determined using multiplex quantitative RT-PCR [93] or RNA sequencing assays, NGS. These techniques may also allow the assessment of heterogeneity between single CTCs within the same patient [94]. The sensitivity of NGS-based technologies is lower than that of PCR-based technologies and inversely proportional to the number of sites analyzed, with the total exome sequence (WES) having the lowest sensitivity [95]. On the other hand, compared to ddPCR, NGS had a higher sensitivity for individual nucleotide variants, indels and selected rearrangements and has been shown to have a positive percentage agreement of 95% and a positive predictive value of 100% [96].

On the other hand, immunophenotyping with antibodies to proteins of interest (proliferation or apoptosis markers) is the most commonly used approach to CTC characterization but is currently limited to a few proteins of interest (beyond those required for the enrichment and detection). In many studies immunophenotyping has been used to confirm the epithelial [97] or mesenchymal [98] nature of the suspected circulating cells (Table 1). However, even among the epithelial markers typically used to conceive CTCs, such as EpCAM or cytokeratins, there is no consensus on specific markers that can more effectively identify clinically relevant CTCs.

A micro-fluid single cell western blot (scWB) technology has also been developed for proteomic CTC phenotyping but is limited to evaluating only eight proteins [99].The rare cell scWB quantifies multiple surface and intracellular signaling proteins in each individual CTC, allowing estimates of the variation in biological protein expression between CTCs. This method is compatible with well-established CTC isolation tools and can successfully analyze CTC populations with just two primary cells. The monitoring of multiple regulated proteins in blood derived CTC may provide information about the treatment options to maximize the benefit for each specific patient at each specific time point [100].

With in vitro cultures of CTCs, in addition to transient expansion, some groups have been able to create permanent CTC cell lines obtained from patients with advanced-stage diseases. However, these cell lines have phenotypes that reflect those of cells in tumor tissue samples from patient donors, but they also have a special molecular signature that reflects the metastasizing capacity generally attributed to CTCs. In practice, cell lines derived from CTCs have germinality, a specific DNA repair phenotype and a high metabolic rate [101].These cell lines can also be used to test drugs in prospective discovery projects, but the process of determining these cell lines is not yet fast enough and CTCs capable of metastasis are a rare subset of the cell population, thus limiting the usefulness of this approach for decision-making in clinical practice.

Furthermore, short-term CTC cultures could provide information quickly enough to potentially inform treatment decisions for the donor patient. They could also reveal new pathways specific to metastasis-causing CTCs and, therefore, new targets for drugs that specifically eliminate this more aggressive subset of CTCs. The evolution of CTCs presents another challenge for the development of cell lines that accurately reflect the disease, and the creation of multiple cell lines using CTCs isolated from sequential blood samples collected during disease and treatment can provide unique information [102].

Finally, CTCs can also be characterized through functional studies in patient-derived xenograft models (PDX) which can result in revealing the properties of these cells that are required for the transition to secondary sites and/or the outgrowth of diffuse cancer cells (DTCs) to form apparent metastases. In addition, these PDX models can be used to test drugs that may be interesting candidates for anticancer therapy [103] (Figure 1). The disadvantage of this method, however, is that the development of PDX models usually takes several months and the rate of successful CTC integration is generally very low due to the requirement of a large number of CTCs, which generally excludes the use of such models in making treatment decisions for individual patients. However, these models appear to recap the molecular and cellular characteristics of parent tumors as well as the response to chemotherapy [86,104].

## 3. ctDNA

Cell-free DNA (cfDNA) from cancer cells, known as circulating tumor DNA (ctDNA), can be tracked in the plasma of cancer patients. Since the first reporting of identical DNA mutations in plasma compared to a patient’s tumor, ctDNA has been investigated as a tool for diagnosis, detection, prognosis, treatment selection and monitoring [105]. Both the amount [106] and integrity [107] of circulating cfDNA can be used to distinguish between cancer patients and healthy individuals. Overall levels of cfDNA tend to be higher in cancer patients than in healthy individuals [108,109,110] and appear to increase with stage [111] and metastasis [112]. The increased concentration of cfDNA in these patients is believed to reflect the additional release of genetic material from tumor cells, but it could also be a result of defective clearance of ctDNA from phagocytes [113]. However, high levels of cfDNA are not specific to cancer and have been identified in other pathological and non-pathological conditions, such as exercise, trauma, and surgery, that may interfere with their immediate application for a cancer diagnosis [114] (Figure 1).

### 3.1. Detection of ctDNA and Current Limitations

Currently, highly sensitive and specific methods for the detection of ctDNA have been developed. Technologies leading to the detection of ctDNA can be separated into two main categories: (a) targeted techniques designed to detect mutations in a collection of predetermined genes, and (b) untargeted methods that attempt to screen the entire genome, such as whole-genome sequencing, exome sequencing, or array comparative genomic hybridization [115]. Allele-specific PCR was the first approach used in ctDNA detection, and a quantitative PCR variant (qPCR) of this technique is currently being adopted by the *EGFR* cobas^®^ test [116]. Given that the proportion of ctDNA in total cfDNA is typically very low, frequently 0.01% [117], more sensitive technologies, such as digital PCR (dPCR) [118], droplet digital PCR (ddPCR) [119], and beads, emulsion, amplification, magnetics (BEAMing) [120], have been developed and successfully used for ctDNA analysis (Table 2). The inadequate multiplexing capability of PCR-based assays prevents them from analyzing more than a few loci concurrently, despite their high sensitivity, speed, and affordability. These techniques have ctDNA detection limits of 0.01%, making them more sensitive than non-targeted sequencing procedures. However, the need for extensive prior knowledge of the mutational spectrum of the tumor in the specific patient is a drawback of these methods [115].

The sensitivity of NGS-based technologies is negatively correlated with the number of loci tested, lower than that of PCR-based technologies, and lowest for whole-exome sequencing (WES) (5% mutant allele fraction (MAF)—the percentage of mutant allele in a given locus). Consideration of patient- or cancer-specific gene panels, as in the cancer personalized profiling by deep sequencing (CAPP-Seq) technology [121], or strategies to suppress background noise generated by random errors occurring during library preparation are approaches to improving NGS sensitivity. These methods include attaching distinct molecular identifiers to each template molecule (UMIs). Various NGS technologies, including improved tagged amplicon sequencing (eTAm-SeqTM), utilise these [105] (Table 2). Selective nuclease digestion of DNA that has not undergone mutation is another method to boost sensitivity. This method raises MAF and has made it possible to detect mutations down to 0.00003% MAF [108].

Despite its potential, using ctDNA as a liquid biopsy has a number of limitations. The sensitivity of detection is a serious concern, particularly in early cancer detection, where the low amount of ctDNA may result in a MAF lower than the detection limit of existing techniques (Figure 1). Other body fluids sampled near the putative site of the tumor can increase the detection rate, at least in individuals at risk due to hereditary predisposition for example. This is primarily because, particularly in the early stages, proximal body fluid may contain a higher concentration of tumor-derived DNA than blood [122]. Another issue in early detection is the predictive value of single or small groups of mutations, because cancer-associated mutations can be found in healthy people’s plasma as a result of clonal hematopoiesis. The CancerSEEK platform, which associates the analysis of eight tumor-derived proteins with ctDNA mutation profiling and has a specificity of >99%, is one approach to overcoming this challenge [123]. Another barrier to the widespread use of ctDNA analysis is the lack of standardized protocols for preanalytical sample preparation and ctDNA purification. Current procedures are complicated and have the potential to cause ctDNA degradation and blood cell lysis [124]. It is desirable to have a platform that allows for the quick, single-step purification of ctDNA from blood and lab-on-a-chip systems have the potential to meet this need [125].

**Table 2 cancers-15-01579-t002:** Clinical applications of ctDNA, ctRNA and EVs.

	Cancer	Markers	Technique	References
ctDNA	Breast	*ERBB2*, *BRCA1*, *TP53*, *PIK3CA*	NGS Tam-Seq	[126] [127]
	Colorectal	*BRAF*, *KRAS*, *APC*, *TP53*, *CEA*, and *SEPTIN9*	Sanger Sequencing, Epi proColon Assay	[126]
	Gastric	*ERBB2*, *FGFR1*, *CDH1*, *PIK3CA*, *MET*, *KDR*, *TP53*, and *TFDP-1*	qPCR CellSearch	[126] [128]
	Gliomas	*IDH1*, *EGFR*, *KRAS*, *MGMT*	qRT-PCR, ddPCR, BEAMing	[129]
	Head and neck	*TP53*, *PIK3CA*, *NOTCH1*, *FBXW7*, *CDKN2A*, *NRAS* and *HRAS*	NGS	[130,131]
	Hepatocellural	*TP53*, *CTNNB1*, *PTEN*, *CDKN2*, *ARID1A*, *MET*, *CDK6*, *EGFR*, *MYC*, *BRAF*, *RAF1*, *FGFR1*, *CCNE1*, *PIK3CA* and *ERBB2*	Sanger Sequencing, Mass Spectrometry, qPCR	[126]
	Lung	*EGFR*, *ALK*, *BRAF*, *KRAS*, *ERBB2*, *PIK3CA*, *FGFR1*, *KRAS*, *ROS1*, *MET*, *RET* and *TP53*	NGS ddPCR	[132,133,134]
	Pancreatic	*KRAS*, *BRAF*	NGS ddPCR WES	[131,135,136]
	Prostate	*TP53*, *RB1*, *PTEN*, *AR*, *FOXA1*, *MYC*, *ERG*, *PIK3CA*, and *WNT1*	CellSearch	[137]
RNA	Colorectal	miR-548c-5p, miR-21, CRNDE-h, lncRNA GAS5, miR-19, miR-221, lncRNA 91H, miR-23a, miR-1224-5p, miR-6803, Let-7a, miR-1229	qRT-PCR	[138,139,140,141]
	Gastric	lncUEGC1, lncUEGC2, HOTTIP, ZFAS1, miR-423-5p, miR-451, miR23b	qRT-PCR	[142,143,144,145]
	Pancreatic	MiR-125b-5p, miR-21, circPDE8A, miR-451a, miR-191, miR-17-5p	qRT-PCR	[146,147,148]
	Liver	hnRNPH1, LINC00161, LINC00635, TERT, miR-638, miR-125b, miR-93	qRT-PCR	[149,150,151]
	Laryngeal	HOTAIR	qRT-PCR	[152]
	Prostate	MiR-1290, miR-375, miR-125, miR-19b, SAP30L-AS1, SChLAP1, LincRNA-p21	qRT-PCR	[153,154]
	Ovarian	miR-200a, miR-200b, miR-200c, miR-21, miR-100, miR-326	PCR, qRT-PCR	[155]
	Lung	miR-451a, miR-23b-3p, miR-21-5p, miR-10b-5p, MALAT-1	qRT-PCR	[156,157]
	Multiple Myeloma	let-7b, let-7e, miR-106a, miR-106b, miR-155, miR-16, miR-17, miR-18a, miR-20a	qRT-PCR	[158]
	Glioma	miR-301a	qRT-PCR	[159]
	Glioblastoma	RNU6	qRT-PCR	[160]
EVs proteins	Colorectal	CPNE3, GPC1	ELISA	[161,162]
	Pancreatic	GPC1	Flow cytometry	[163]
	Lung	CD171, 14-3-3ζ, Flotilin 1, HER3, GRP78	ELISA	[164,165]
	Ovarian	ephrin A2	ELISA	[166]
	Prostate	ADIRF, TMEM256	Mass spectrometry	[167]
	Melanoma	exo-MIA, exo-5100B	ELISA	[168]
	Bladder	TACSTD2	ELISA	[169]
	Breast	ERBB2, BCRP, Fibronectin, Periostin, Del-1	Flow cytometry ELISA	[170,171,172]

### 3.2. Clinical Applications

Five tests have been approved by the FDA since the discovery of cfDNA. These tests include finding point mutations in cancer-related genes like *KRAS*, *EGFR* and *PIK3CA*, as well as assessing tumor mutation burden (TMB), microsatellite instability, *ALK* rearrangement, insertions and deletions, and methylation patterns [173] (Table 2). The results of these tests may have an immediate impact on the patient’s treatment. Early cancer detection, improved cancer staging, early detection of relapse, real-time monitoring of therapeutic efficacy, and detection of therapeutic targets and resistance mechanisms are all current clinical applications [174].

ctDNA analysis can provide both qualitative and quantitative information. The MAF measurement provides quantitative information and is a reflection of tumor burden. It is used to detect minimal residual disease (MRD) and occult metastases [175], as well as to monitor treatment response and therapeutic effectiveness [176]. Because ctDNA has a short half-life (2.5 h), ctDNA levels provide a ‘real-time’ snapshot of tumor bulk. The presence of ctDNA after treatment is a highly sensitive and specific predictor of relapse [177]. The profiling of mutations, amplifications, deletions and translocations in ctDNA can provide qualitative information, allowing the identification of genetic alterations associated with response and thus supporting decision-making for personalized management. Other qualitative information obtained from ctDNA analysis includes methylation status [178] (Figure 1).

### 3.3. ctDNA in Other Biofluids

Other biofluids, besides blood, have been shown to contain ctDNA including urine, cerebrospinal fluid (CSF) and gastric washes. Depending on the type of cancer, tumors may come into closer contact with different fluids, resulting in higher ctDNA concentrations than blood [179]. CSF-derived ctDNA is particularly easy to investigate because it is not diluted by the normal DNA found in blood. A few studies have looked at CSF and paired plasma, tumor tissue from patients with central nervous system tumors (glioblastoma and medulloblastoma), as well as brain metastases from lung or breast cancer [180]. In patients with head and neck squamous-cell carcinoma, the presence of saliva-derived ctDNA has been used to detect HPV and genomic point mutations. Saliva ctDNA was found to be enriched for ctDNA from the oral cavity, whereas plasma ctDNA was found to be enriched for tumor DNA from other sites [130].Tumor-specific genomic and epigenomic alterations in urine-derived ctDNA have been observed in patients with urological, prostate, NSCLC, CRC, pancreatic cancer and other cancers. However, assessing urine-derived ctDNA is more difficult due to the massive amount of normal DNA constantly released by urinary epithelial cells [181].

## 4. ctRNA

The percentage of circulating cell-free RNA derived from cancer cells is known as ctRNA. In comparison to DNA, RNA is a rather unstable molecule, with a naked half-life in plasma of 15 s [182] (Figure 1). Its interaction with proteins [183], proteolipid complexes and EVs increases its stability [184].

### Clinical Applications and Limitations

Almost all known types of RNA have been detected in systemic circulation, and each has the potential to act as a cancer biomarker to some extent. ctRNA, like other components of the tumor circulome, provides both quantitative and qualitative information. In reality, whereas coding and noncoding RNA expression patterns are the most relevant source of information, the discovery of tumor-specific fusion transcripts or alternative splice events is also achievable [185] (Figure 1). The most important ctRNAs that might be used as biomarkers include mRNAs, miRNAs and long noncoding RNAs (lncRNAs) (Table 2). Their study is carried out using techniques ranging from qRT-PCR or dPCR-based evaluation of single or small panels of RNAs to RNA-Seq-based complete characterization of RNA (particularly miRNA) signatures [186].

Several miRNA levels are often changed in cancer patients, allowing for the identification of miRNA signatures with diagnostic and prognostic value. Tumors and their microenvironments produce miRNAs that are released in the circulation as ribonucleoprotein complexes or as EVs [187]. Circulating miRNA patterns appear to be consistent with tumor tissue profiles [188]. EV-incorporated miRNAs, on the other hand, appear to comprise just a tiny percentage of the miRNAs in circulation and to have different diagnostic performance [189]. Plasma exosomal miR-196a and miR-1246 levels have the potential for early pancreatic cancer detection [190], and panels of miRNAs have been demonstrated to be valid biomarkers for lung cancer diagnosis [191] or prognosis [156]. A serum exosomal miRNA profile was recently demonstrated to be a novel approach for the differential diagnosis of gliomas [192] (Table 2).

Exosomal mRNA has been utilized to explore the mutational status of *KRAS* and *BRAF* in CRC patients [193], and exosomal EGFRvIII mRNA has the potential to be employed in the diagnosis of EGFRvIII-positive high-grade gliomas [194]. Numerous lung-cancer-related gene fusions have been found in both vesicular and nonvesicular mRNA and have potential as biomarkers [195] (Table 2).

LncRNAs are a new and potential source of RNA biomarkers. Plasma exosome LINC00152 levels, for example, have been associated with gastric cancer, and the combination of two mRNAs and one lncRNA in serum exosomes has CRC diagnostic potential [196]. Furthermore, serum exosomal HOTAIR lncRNA can be used to help in the diagnosis and prognosis of glioblastoma multiforme [197]. Recently, a panel of five circulating lncRNAs were investigated as potential diagnostic biomarkers for gastric cancer [198] (Table 2).

The most significant obstacles to the clinical use of ctRNAs concern the preanalytical and analytical phases. Although circulating RNAs are protected by their connection with various molecules and structures, they are unstable in plasma when held at 4 °C and are restricted by extraction speed. Furthermore, different extraction techniques provide varying recovery rates, and there is presently no agreement on the best extraction protocol [199] (Figure 1).

## 5. Extracellular Vesicles (EVs)

EVs are membrane particles that are produced by all cell types under healthy and pathological situations as well as in response to various stimuli such as proteases, ADP, thrombin, inflammatory cytokines, growth factors, biomechanical shear and stress inducers, and apoptotic signals [200]. They may be present in nearly every physiological fluid, particularly blood. EVs are classified into two groups based on their biogenesis, composition and secretory pathways: microvesicles and exosomes [201]. Exosomes belong to a large class of EVs with a diameter ranging from 40 nm to 160 nm that are created by the inward budding of the limited multivesicular body (MVB) membrane, which is generated constitutively from late endosomes. Intraluminal vesicles (ILVs) occur within large MVBs as a result of late endosomal membrane invagination. Some proteins are integrated into the invaginating membrane during this process, whereas cytosolic components are absorbed and confined inside the ILVs. As ILVs fuse with the plasma membrane, the majority of them are discharged into the extracellular space as “exosomes”. Research suggests that the endosomal sorting complex required for transport (ESCRT) function is essential for the production of ILVs [202]. Notably, new research suggests that an alternate method for sorting exosomal cargo into MVBs that is ESCRT-independent appears to rely on raft-based microdomains for lateral cargo segregation inside the endosomal membrane. These microdomains are hypothesized to be rich in sphingomyelinases, which can be used to create ceramides by hydrolytic removal of the phosphocholine moiety. Ceramides are known to cause lateral phase separation and microdomain coalescence in model membranes. Moreover, ceramide’s cone-shaped structure may generate spontaneous negative curvature of the endosomal membrane, enhancing domain-induced budding. As a result, this ceramide-dependent method highlights the importance of exosomal lipids in exosome synthesis [203]. The mitogen-activated protein kinase pathway, which is up-regulated in most tumor cells, is considered to be involved in the active shedding of vesicles from tumor cells [132]. They are distinguished by characteristics such as CD9, CD63, CD81, ALIX and heat shock protein 70 (HSP70), which aid in their collection and enrichment [204]. Exosomes can mediate cell communication under healthy and pathological settings by transporting particular cargos (nucleic acid or protein) [205]. Exosomes, as evidenced by growing studies, play an important role in carcinogenesis, tumor development, metastasis and medication resistance [206]. The genetic makeup of the parent tumor cells is congruent with the cargos of tumor-derived exosomes [207]. As a result, exosomes and their transported cargos have progressively come to be recognized as new biomarkers for cancer diagnosis and prognosis prediction. Exosomes are also stable in circulation and can preserve their cargos from degradation [208].

### 5.1. Isolation of EVs

The absence of established guidelines for sample handling and EV isolation and analysis, which could affect reproducibility in the clinical area, is an important limitation to the clinical application of EVs as liquid biopsies [209] (Figure 1). Mainstream EV isolation methods utilise physiological (density and size) and biological (expression of surface markers) characteristics [210].

Ultracentrifugation (UC) is now recognized as the “gold standard” approach for separating and concentrating exosomes from other components depending on densities. Protein contamination can be reduced by UC. Though, it has a limited throughput and may separate other particles of comparable size [211]. While the throughput is still limited, utilizing density gradient centrifugation can overcome the impurity of the UC technique [212]. Although commonly utilized, these procedures are costly, time-consuming, and do not guarantee pure yields, frequently resulting in a trade-off between purity and recovery.

Filtration and size-exclusion chromatography (SEC) are two size-based approaches. Filtration can provide high yields and purity, but it is restricted by EV adhesion to filters and vesicle destruction caused by high pressure. Furthermore, its poor resolution is restricted by the presence of additional contamination such as virus and lipoprotein particles [213]. When compared to ultracentrifugation, SEC offers enhanced EV recovery [214].

Immunoaffinity-based separation techniques involve antibodies to target particular surface antigens of exosomes, which can greatly boost exosome purity and reduce isolation time [215]. Antibodies are often immobilized in ELISA plates or magnetic beads. However, it is expensive and occasionally afflicted by nonspecific antibody binding [216].

Polymer precipitation is another popular technique of isolation, particularly for exosomes. The use of polymers such as polyethylene glycol (PEG) to limit the solubility of EVs in order to precipitate them using quick low-speed centrifugation is used in this procedure. This approach has a low purity despite providing good recovery rates [217].

Electric fields are being used in new methods for EV isolation. Lewis et al. created an alternating-current electrokinetic (ACE) chip that can catch exosomes from the entire blood sample and perform in situ immunofluorescent analysis in 30 min. The scientists verified this chip by evaluating GPC-1 and CD63 levels as PDAC diagnostic indicators [218].

Finally, microfluidics is a promising area of prospective innovative techniques to EV isolation. The existing microfluidic techniques are based on EV characteristics including nanoscale size-based filtering [219], antibody-functionalized microfluidic channels [220] and spiral inertial microfluidic devices [221].

Exosomes must be measured and examined once they have been extracted. Exosomes are widely quantified using ELISA, fluorescence activated cell sorting (FACS), and nanoparticle tracking analysis (NTA). ELISA can capture certain proteins and generate a color change that is proportional to the concentration of the target protein. CD9, CD63 and CD81 have been identified as the most often utilized exosome-specific markers in the ELISA approach for exosome quantification [222].

Exosome-specific markers can also be employed in FACS to quantify and sort exosomes. However, FACS needs a somewhat sophisticated setup and costly equipment, making it unsuitable for clinical use. Another disadvantage of FACS is the lack of consistency in results due to the various optical and laser settings used to detect exosomes [223].

Another fluorescent-based approach for measuring and sorting exosomes is NTA. The idea is to use a laser beam to follow the movement of exosomes. NTA can identify smaller exosomes than FACS; however, it cannot be used in clinical settings due to the lengthy processing time [224]. As a result, various innovative ways for detecting and measuring exosomes arise that are more cost-effective and efficient. Lv et al., for example, coated nanoellipsoids with antiCD63 antibody as the substrate of localized surface plasmon resonance biosensors. The peak wavelength can be used to calculate exosome concentration. When compared to ELISA, this type of biosensor takes a fifth of the sample amount but can cut processing time in half. Furthermore, it is inexpensive, which makes it suitable for clinical use [225].

### 5.2. Clinical Applications

The molecular contents carried by EVs can be regarded as a molecular fingerprint of the cell of origin, making them suitable as cancer biomarkers. When compared to CTCs, EVs are frequently produced and liberated in higher numbers [226]. Similarly, the vesicular cargo’s stability is maintained by an outer protective lipid membrane. EVs can provide both quantitative and qualitative data. Quantitative data such as EV levels can reveal the existence of malignant etiology and tumor density. The most easily accessible qualitative information is gained by the molecular characterization of EV components, including nucleic acids and proteins. The RNA composition of EVs, including both coding and noncoding (nc)RNAs, has received a lot of attention. Proteins are carried by EVs in their lumen and membrane, and multiple studies have been published indicating the importance of EV proteins as potential cancer biomarkers [227] (Table 2).

## 6. Proteomics

From a proteogenomic approach, evaluating the proteome is more technically and conceptually rigorous than analyzing the genome. To begin, the proteome is projected to have about one million diverse proteoforms via multiple epigenetic controls, variable RNA splicing and PTM, as opposed to a total of 22,000 to 25,000 protein-translatable genes inside the human genome. Furthermore, the dynamic range of proteins in cells or body fluids can reach up to 12 logs of size [228]. Finally, the proteome undergoes continual and fast shifts in protein quantities and/or alterations in response to a variety of stimuli. While it is impossible to test the identical proteome twice, the genome is generally stable, with gradual continuous alterations. Because of these difficulties, proteomics typically comes behind genetics in many applications [229]. However, as proteins are the primary mediators of most biological activities and the direct drug targets in the majority of existing cancer treatments, high-dimensional proteomic data are anticipated to yield unparalleled insights to contribute to the identification and practical use of new biomarkers (Figure 1). High-plex proteomics technologies that are applied in cancer liquid biopsy include mass spectrometry (MS), antibody/antigen arrays, aptamer-based assays, proximity extension assay (PEA) and reverse phase protein arrays (RPPA) [230].

### 6.1. Mass Spectrometry

MS-based proteomics constitute an effective method for cancer biomarker profiling in the context of various body fluids, with an emphasis on serum/plasma and urine. With technological and scientific advancements, current MS mostly employs purpose-designed sample preparation in conjunction with liquid chromatography (LC) prior to peptide ionization and tandem MS scans in liquid biopsy screening [231]. The ability to conduct non-hypothesis-driven proteome research (total proteins and modified forms) is a fundamental feature of MS for cancer liquid biopsy, making it a preferable technique at the early biomarker identification stage. Nowadays, for clinical proteomic analysis, a few hundred to over a thousand proteins may be described in an untargeted MS run in serum or plasma, but urine-based MS profiling can accomplish several thousand targets concurrently due to its considerably less complicated protein composition [232]. The main task in blood-based proteomics is to decrease noise or false discovery rates due to the huge dynamic range of blood protein concentration as well as pre-analytical fluctuations [233]. MS-based liquid biopsies, though, have been used in a variety of malignancies, including lung, breast, colorectal, ovarian, gastric, pancreatic, prostate, cervical, lymphoma and so on [234,235,236,237] (Table 2).

### 6.2. Antibody/Antigen Arrays

Immobilizing particular antibodies onto modified planar substrates by covalent binding, affinity binding or physical trapping is a common scientific technique. Samples are typically tagged with fluorescent, chemiluminescent or oligo-coupled tags in high-plex (usually several hundred targets) profiling to allow for varied signal amplification and detection. This approach is capable of characterizing over a thousand proteins or modified proteoforms with low immunogenic cross-relativity caused by antibody reaction mixtures [238]. Because most TAP are low abundant cellular efflux, including hormones, cytokines, chemokines, intracellular signaling components and post-translational alterations, antibody arrays are especially effective for serological analysis [239]. Nonetheless, because of its inadequate quantification due to restricted dynamic ranges and signal saturation, sample labeling need, and inter-assay heterogeneity, it occupies a tiny methodological area for biofluid-based proteomic screening. Another high-throughput discovery in proteomics field is antigen arrays, also known as functional protein arrays [240]. They begin with the deposition of ectopically produced proteins/peptides with broad proteome coverage in the desired species which act as baits to collect analytes of interest inside the flowthrough. Protein interactions with proteins (protein PTMs), lipids, cells, tiny molecules, nucleic acids and antibodies may all be studied theoretically. In this regard, serological autoantibodies (AAbs) constitute a hotspot for cancer biomarker profiling [241].

### 6.3. Aptamer-Based Assays

Aptamers are short single-stranded DNA, RNA or peptides that bind to cognate protein targets in natural states with high affinity and specificity after folding into specified tertiary structures [242]. In the case of the slow off-rate modified aptamers (SOMA) scan assay, binding molecules (SOMAmers) are attached to photocleavable linkers and fluorescent labels, and those nucleic acid structures are then used to capture proteins of interest, followed by biotin-mediated purification, oligo release via ultraviolet (UV)-based cleavage, and biotin tagging of bound proteins. The protein-bound SOMAmers are then eluted and quantified using traditional DNA hybridization methods, representing the protein abundance in the system [243]. Aptamers are more beneficial than antibodies because they have stronger affinity and specificity, and they can be easily produced and chosen in vitro with little batch-to-batch variation, giving a cost-effective means to scale up their multiplexity [244]. Regardless of the fact that there are already over 7000 protein-specific aptamers available for commercial assay services, one constraint is the difficulty in creating high-quality aptamers for new targets. Aptamers are still scarce in the research community compared to antibodies [245].

### 6.4. Proximity Extension Assay (PEA)

Its wide dynamic range (scan 10 logs) and small sample size make PEA an ideal method for serological analysis. Multiple antibody pairs for proteins of interest are pooled in PEA. Each antibody in a pair is tagged with complementary DNA oligo sequences to enable for high-fidelity discriminative hybridization, which occurs only when real antibody pairs are brought together by binding to the target proteins. Following that, the proximity reactions proceed through a dilution phase, which replaces the washes used in typical sandwich immune tests. Oligonucleotides on pairs of antibodies that remain in close proximity due to binding the same protein molecule can subsequently be ligated (proximity ligation assay) or polymerized (DNA polymerization assay) (proximity extension assay). PCR is used to amplify the resulting double-stranded DNA sequences. The ligation or polymerization procedures produce amplifiable reporter DNA strands for sensitive readout using techniques such as real-time PCR or next-generation sequencing [246]. The most developed PEA assay offers standard measurement coverage of 3072 targets and avoids the cross-reactivity problem that multiplexed immunoassays generate [247].

### 6.5. Reverse Phase Protein Arrays (RPPA)

RPPA is an open-source platform that may be constructed in a variety of ways. In a typical RPPA system, completely denatured protein lysates are immobilized onto solid substrates, often by a dilution series, and this procedure can be repeated to probe any number of targets (up to 500 targets). Sample-containing slides are probed with highly specialized antibodies that have been pre-validated for RPPA use, and quantifiable signals are collected via colorimetric amplification or fluorescence analysis. Due to its nature of measuring all samples in one test cycle, which typically runs from a few hundreds to over a thousand samples, RPPA is extremely resilient in parallel to big sample profiling [248]. RPPA necessitates a complicated experimental procedure that includes critical steps such as array printing, numerous phases of immunostaining and signal amplification, high-resolution data outputs, and custom data compilation and analysis [249].

## 7. Metabolomics

Metabolomics is regarded as a potent high-throughput tool for detecting low molecular weight compounds in biological samples such as blood, urine, bile, ascites and tissue. So far, it has contributed to the clarification of biochemical processes involved in numerous human malignancies, while also providing a unique opportunity to identify novel biomarkers and carcinogenesis drivers in this field [250]. Cancer metabolomics, for example, has shown an upregulation in glycolysis, glutaminolysis, lipid metabolism, mitochondrial biogenesis and the pentose phosphate pathway, among other biosynthetic and bioenergetic pathways [251].The most forefront example of metabolomics supporting precision medicine is the use of a metabolomic method to categorize malignancies in order to later build personalized medicines [252].

Another main application area for metabolomics is the development of cancer medicines. Cancer immunotherapy, for example, has lately altered the paradigm in a number of solid and hematologic cancers. However, in a considerable proportion of instances the responses are limited, with cancers acquiring inherent or acquired resistance to checkpoint inhibition [253]. Certain immune-sensitive cancers develop immunity, resulting in tumor growth and disease progression. The tumor microenvironment is the most important contributor to immune resistance [254]. By modifying immune metabolism and reprogramming immune cells, nutrient shortage, hypoxia, acidity and the release of numerous inflammatory markers all contribute to pro- or anti-inflammatory phenotypes [255].

A wide range of matrices can be investigated from all available tissues and body fluids, such as plasma, serum, cerebrospinal fluid (CSF), saliva, feces, pus, cervicovaginal secretions and urine. Because of the chemical complexity of the metabolome, the dynamic range of metabolites, fluctuating quantities and the hard simultaneous quantification within complex mixtures, identifying a metabolome as a lengthy metabolite list by accurate spectrometry-quantification is complex [256]. However, the Metabolomics Society has set reporting criteria for biospecimen source, collection and processing details [257].

Nuclear magnetic resonance (NMR), gas chromatography-mass spectrometry (GC-MS), liquid chromatography-mass spectrometry (LC-MS), capillary electrophoresis-mass spectrometry (CE-MS) or other combinations of these analytical methods can all be used. NMR can be used to identify metabolic signatures or biomarkers associated with homeostasis disorders. In cancer research, mass spectrometry imaging in combination with the rest of the methods can contribute to three possible applications: (i) establishing a chemical and morphological mapping of regions of interest to identify next-generation prognostic and therapeutic biomarkers, (ii) evaluating the molecular efficacy of chemotherapeutic agents, and (iii) classifying tissue types based on molecular patterns to recognize their pathways and therapeutic prognoses [258].

## 8. Future Directions

Liquid biopsy is a very informative and noninvasive method for the care of cancer patients because it offers information on the molecular properties of the tumor in real time, recapitulating the entire tumor complexity (Figure 1). Because blood collection may be done frequently, liquid biopsy is also highly essential in helping us understand how the tumor changes as it advances. This method becomes more useful when traditional biopsies are not possible. In spite of the therapeutic importance of liquid biopsy, which has been demonstrated in several trials in various forms of cancer, its clinical value is just now beginning to penetrate the clinic. Based on ctDNA analysis, the FDA has authorized liquid biopsy NGS companion diagnostic assays for numerous malignancies and biomarkers (Table 2). It is worth mentioning at this time that ctDNA analysis is progressing faster than CTC analysis and has already achieved significant clinical use in standard practice [259].

It is critical for the success of downstream applications to ensure that ctDNA samples are of appropriate number and quality. Contamination of samples with genomic DNA must be avoided for this purpose, for example, by employing white blood cell stabilizers. It is also preferential to isolate ctDNA from plasma samples rather than serum, as this avoids the release of cellular DNA from lysing cells during the clotting process. Furthermore, due to the low concentrations, extraction procedures must provide high ctDNA yields [260]. Because of the tiny amount and proportion of ctDNA in circulation, extremely sensitive detection methods such as droplet digital Polymerase Chain Reaction (ddPCR), Next Generation Sequencing (NGS) or BEAMing (beads, emulsion, amplification and magnetics) must be used [261]. Nevertheless, because of temporal variability, targeted sequencing restricts treatment response monitoring and identification of resistance mutations, highlighting the necessity for larger panels to assess ctDNA during follow-up, which may impair detection sensitivity [262]. In clinical practice, ctDNA profiling is still complicated and costly [263].

Furthermore, the isolation of CTCs, which are relatively scarce in circulation, is challenging and expensive as well [264]. CTC molecular characterization, on the other hand, can give extra information (Figure 1). Single CTC genomic analysis can show intrapatient heterogeneity, which may explain therapy resistance. Additionally, transcriptional plasticity may be a significant driver of cancer therapeutic resistance, and CTCs may be probed at the RNA and protein levels. Transcriptional analysis of CTCs may be able to predict which organ site will be colonized in the future. More specifically, different organ microenvironments can collect different types of tumor cells and induce different transcriptional activities as a result of crosstalk between tumor cells and surrounding organ cells. Single-cell CTC analysis might reveal intrapatient heterogeneity. Finally, CTC-derived cell lines or xenografts can be employed as novel models for screening tests, opening up a new route for functional investigation [102].

Exosomal nucleic acid and protein have been implicated in carcinogenesis and tumor progression in a rising number of studies in recent years, indicating that they might be used as a diagnostic or prognostic biomarker. However, research on exosomal lipids and metabolites as diagnostic or prognostic indicators is inadequate [205]. Despite the various advantages, the use of exosomes as cancer biomarkers is fraught with difficulties. To begin, conventional methods for separating and enriching exosomes have limited throughput and purity. There are significant differences in the procedures used to isolate exosomes (Figure 1). Thus, the current aim in this research is to enhance the exosome isolation and enrichment procedure, create more efficient characterization approaches and finally establish a standard exosome-based method. Furthermore, it is questionable if the relatively low quantity of exosomes in biofluids is adequate to detect minute abnormalities which are frequently overlooked in clinical detection. Evaluation of global abnormalities such as chromosomal instability may help to solve this problem [265]. Furthermore, the demonstration of the superiority of exosomes as a liquid biopsy is based on limited patient cohorts and lacks a clear therapeutic advantage [266]. As a result, it is critical to develop accurate exosomal biomarkers in large-scale samples for early-stage cancer detection and prognosis prediction that can be modified for therapeutic use.

The advancement of the technological aspect of proteomics is one potential direction. Improving detection resolution, standardizing procedures and increasing high-quality antibodies with high sample throughput can improve overall detection accuracy, particularly during the early stages of discovery. This is due to the fact that most organ-specific biomarkers in the secretome are present in extremely low quantities and have yet to be found. Given the absence of a so-called ideal technology, balancing the benefits and drawbacks of multiple technologies throughout the development phases is critical [267]. This may be shown in recent research that employed MS or aptamers in conjunction with PEA to uncover cancer biomarkers [268]. Finally, as with liquid biopsy, single-cell proteomics is spreading across all domains of cancer biomarker research. MS has already opened the path for single-cell proteomics using flow-cytometry cell sorters and high-resolution TIMS-TOF [269]. Surface protein phenotypes and single-cell secretomes are both hotspots for finding novel biomarkers in liquid biopsy, notably in cancer immunotherapy [270].

## 9. Conclusions

The study of tumor genetic changes from tissue samples is one of the current criteria for patient classification and therapy selection. Tissue biopsies, while undeniable in their importance, have significant limitations in that they are very invasive procedures that fail to capture tumor clonal heterogeneity. Liquid biopsies, which include the examination of circulating tumor-derived components such as CTCs, ctDNA or ctRNA, and EVCs, are gaining popularity as a promising treatment option. The tumor circulome contains many kinds of tumor-derived biological components. Novel methods are being developed to improve tumor circulome analysis, with the goal of fully investigating the intricacy of the information obtained from a single blood sample.

The potential of liquid biopsies and the advent of new technology enables researchers to define each individual component of the tumor circulome with greater precision. Liquid biopsies are being hailed as a game-changing technique in customized cancer care. Because of advancements in both omics’ technologies and the associated artificial intelligence elaboration of the data, liquid biopsies can overcome many limitations of tissue biopsies and can capture tumor heterogeneity in general, but mostly they can capture tumor evolution without being invasive to the patients. This will soon be converted into a more precise prognosis evaluation and the optimum therapy option based on a particular patient’s condition as it progresses, ushering in a genuine precision medicine approach.

Nevertheless, its clinical implementation has been slowed by a number of technological obstacles. As a result, various issues need to be overcome before liquid biopsies may be used in clinical settings. Despite the considerable work that must be done to fully define the future role of liquid biopsies in cancer diagnosis, monitoring and prognosis, the key outcomes published so far indicate the promise of this method in changing current cancer management approaches.

## Figures and Tables

**Figure 1 cancers-15-01579-f001:**
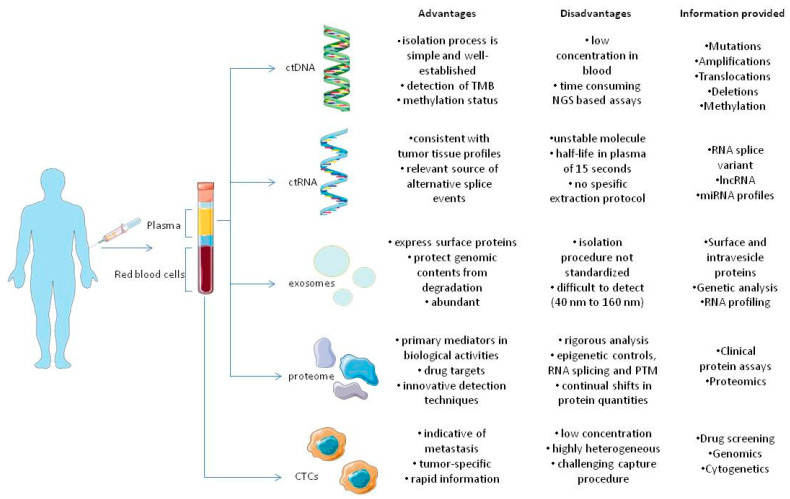
Components of tumor circulome examined in liquid biopsies and their applications. The several analytes extracted from blood provide a wide variety of information regarding tumors. As previously stated, all analytes share different advantages and disadvantages that favor or oppose their usage in clinical settings, in tumor diagnosis, monitoring and therapy [23].

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
