# Peer review of "Liquid Biopsies, Novel Approaches and Future Directions"

_cancers, 2023, doi:10.3390/cancers15051579_

Round 1

Reviewer 1 Report

1)    “Early diagnosis and prognosis are vital to improve patients’ outcome. The golden standard of tumor characterization is tissue biopsy. Amongst the constraints of tissue biopsy …” – the logical transition between these sentences is not clear. Add information and rephrase this.

2)    Figure 1. In the "information provided" part, small drawings are not informative. Delete them and increase the text size.

3)    Сaption of Figure 1. Reference number (262) – place in order the reference numbers in the text, including the figure caption.

4)    «On the other hand, there are some restrictions as for this method. Τhe use of expensive antibodies leads to high detection costs and CTCs cannot be further analyzed in real-time conditions, since the cells are fixed or lysed during the assay process. (50,51)».  There is no need to fix or lyse cells, for example when staining cell surface receptors. So the information is not correct.

5)    Check section numbers and titles. Why “2.4. ctDNA” or “2.10. Extracellular vesicles (EVs)" are located in the section "2. CTCs "? There is a mistake in the title “Extracellural”.

6)    «Though, high levels of ctDNA are not specific to cancer and have been identified in other pathological and non-pathological conditions, such as exercise, trauma, and surgery, that may interfere with their immediate application for a cancer diagnosis (112)» - Perhaps the authors meant cfDNA.

7) “Exosomes belong to a large class of EVs with a diameter ranging from 40 nm to 160 nm that are actively secreted into biofluids by either fusion with the plasma membrane and then exocytosis of multivesicular bodies or direct budding of tiny cytoplasmic protrusions from the cell surface.” – Check this information carefully. The description of exosome biogenesis here is poor and incorrect in some places.

8) Section 3.1. In the last sentence, reference (186) is a review. The authors must cite original research.

9) Section 3.4. I suggest to include a short description of the method.

10) Section 4. There is only 1 reference in the paragraph on cancer immunotherapy. Add links to original research.

Author Response

Dear Sir/Madam 

Thank you very much for your useful comments.

We tried our best to address all your comments/ suggestions. Pleases see attached file.

Best regards

A. Armakolas  

Reviewer 2 Report

The authors provided a well-documented and brilliantly organized overview of the potential impact of liquid biopsy options for tumor management.  In addition to this, the review could represent a really interesting point of view in a field so dynamic and rich in potential future applications.  If the article is well written, the introduction section could be improved by including more detailed argumentations regarding EVs: new technologies for the association of a specific marker with an EVs subtype and the EVs subtype to a particular function and/or group of functions it's becoming vital (PMID: 35141731 and others).  I hope that my comments could be useful and I look forward to reading the revised version of the paper.

Author Response

Dear Sir/ Madam 

Thank you for your useful comments.

We addressed all your suggestions to our best. Please see attached file.

Best regards

A. Armakolas
